# Premixed Calcium Silicate-Based Root Canal Sealer Reinforced with Bioactive Glass Nanoparticles to Improve Biological Properties

**DOI:** 10.3390/pharmaceutics14091903

**Published:** 2022-09-08

**Authors:** Min-Kyung Jung, So-Chung Park, Yu-Jin Kim, Jong-Tae Park, Jonathan C. Knowles, Jeong-Hui Park, Khandmaa Dashnyam, Soo-Kyung Jun, Hae-Hyoung Lee, Jung-Hwan Lee

**Affiliations:** 1Department of Biomaterials Science, College of Dentistry, Dankook University, 119 Dan-dae-ro, Cheonan 31116, Korea; 2Institute of Tissue Regeneration Engineering (ITREN), Dankook University, 119 Dandae-ro, Cheonan 31116, Korea; 3Department of Nanobiomedical Science & BK21 PLUS NBM Global Research Center for Regenerative Medicine, Dankook University, 119 Dandae-ro, Cheonan 31116, Korea; 4Department of Oral Anatomy, Dental College Dankook Institute for Future Science and Emerging Convergence, Dankook University, Cheonan 31116, Korea; 5Department of Bio Health Convergency Open Sharing System, Dankook University, Cheonan 31116, Korea; 6UCL Eastman-Korea Dental Medicine Innovation Centre, Dankook University, 119 Dan-dae-ro, Cheonan 31116, Korea; 7Cell & Matter Institute, Dankook University, Cheonan 31116, Korea; 8Division of Biomaterials and Tissue Engineering, Eastman Dental Institute, Royal Free Hospital, Rowland Hill Street, London NW3 2PF, UK; 9Mechanobiology Dental Medicine Research Center, Cheonan 31116, Korea; 10Department of Dental Hygiene, Hanseo University, 46 Hanseo 1-ro, Seosan 31962, Korea

**Keywords:** bioactive glass nanoparticle, calcium silicate sealer, physicochemical properties, biological properties, *Enterococcus faecalis*, mesenchymal stem cells, osteogenic differentiation

## Abstract

Recently, bioactive glass nanoparticles (BGns) have been acknowledged for their ability to promote interactions with the periapical tissue and enhance tissue regeneration by releasing therapeutic ions. However, there have been no studies on calcium silicate sealers with bioactive glass nanoparticle (BGn) additives. In the present study, a premixed calcium silicate root canal sealer reinforced with BGn (pre-mixed-RCS@BGn) was developed and its physicochemical features and biological effects were analyzed. Three specimens were in the trial: 0%, 0.5%, and 1% bioactive glass nanoparticles (BGns) were gradually added to the premixed type of calcium silicate-based sealer (pre-mixed-RCS). To elucidate the surface properties, scanning electron microscopy, X-ray diffraction, and energy-dispersive spectroscopy were used and flowability, setting time, solubility, and radiopacity were analyzed to evaluate the physical properties. Chemical properties were investigated by water contact angle, pH change, and ion release measurements. The antibacterial effects of the bioactive set sealers were tested with *Enterococcus faecalis* and the viability of human bone marrow-derived mesenchymal stem cells (hMSCs) with this biomaterial was examined. In addition, osteogenic differentiation was highly stimulated, which was confirmed by ALP (Alkaline phosphatase) activity and the ARS (Alizarin red S) staining of hMSCs. The pre-mixed-RCS@BGn satisfied the ISO standards for root canal sealers and maintained antimicrobial activity. Moreover, pre-mixed-RCS@BGn with more BGns turned out to have less cytotoxicity than pre-mixed-RCS without BGns while promoting osteogenic differentiation, mainly due to calcium and silicon ion release. Our results suggest that BGns enhance the biological properties of this calcium silicate-based sealer and that the newly introduced pre-mixed-RCS@BGn has the capability to be applied in dental procedures as a root canal sealer. Further studies focusing more on the biocompatibility of pre-mixed-RCS@BGn should be performed to investigate in vivo systems, including pulp tissue.

## 1. Introduction

Root canal sealers are dental materials used in endodontic treatment to form an impervious seal between the root canal wall and core material (i.e., gutta-percha) [1]. Gutta-percha cones have been universally accepted as a root filling material due to their desirable physical, chemical, and biological properties [2]. However, the antibacterial effects of gutta-percha have been reported to be insignificant even though antimicrobial activity is one of the key characteristics for a root canal sealer [3]. A mineral trioxide aggregate (MTA)-based root canal sealer was subsequently introduced, which is currently used for a variety of applications, including pulp capping, as a dressing for pulpotomies in permanent teeth, and during apexification procedures [4]. There are also some drawbacks of MTA, including its long setting time and mixing procedures [5,6]. To compensate for these drawbacks, a premixed type of MTA sealer (Bright Endo MTA sealer, Genoss, Suwon, Korea) was introduced.

Bright Endo MTA sealer, which consists of 50–70% calcium silicates, 20–30% methyl cellulose, and 10% bismuth oxide, is a premixed type of root canal filling material made of hydraulic and calcium silicates for endodontic treatment. This premixed MTA sealer (pre-mixed-RCS) is known to have an antibacterial effect due to its calcium hydroxide forming ability, biocompatibility, ability to facilitate dental hard tissue formation, and hermetic sealing ability by forming surface interactions between MTA and dentin [7,8,9]. In addition to these properties, an ideal pulp capping material also requires osteoinductive effects, which are significant biological properties in periapical healing performance [10,11,12].

In recent years, bioactive glass (BG) has been applied in clinical trials due to its ability to precipitate hydroxyapatite in aqueous solutions, resulting in tissue regeneration [12,13]. Furthermore, questions have been raised regarding whether the application of nanotechnology to synthesize BG at the nanoscale, which results in a high surface area to volume ratio and improved physicochemical properties, could enhance the physical, chemical, and biological properties of root canal sealers [14,15,16,17]. The physical properties of mineral trioxide aggregates with bioactive glass nanoparticle (BGn) additives have been investigated in only a few studies [18,19], but their biocompatibility has not been examined until recently, and there have not been any studies on BGn additives in root canal sealers.

The aim of this study was to evaluate the physical, chemical, and biological properties of premixed calcium silicate root canal sealer reinforced with BGn (pre-mixed-RCS@BGn). The physical properties, such as flowability, film thickness, setting time, solubility and radiopacity, were measured, and chemical properties, such as pH and ion release, were examined. Then, cellular bioactivity in terms of cytocompatibility and osteogenic differentiation were tested.

## 2. Materials and Methods

### 2.1. Fabrication and Characterization of Bioactive Glass Nanoparticles

For the synthesis of BGns composed of 85% SiO_2_ and 15% CaO, ammonia solution (28%), calcium nitrate tetrahydrate (Ca(NO_3_)_2_·4H_2_O), Pluronic 123 (P123), and tetraethyl orthosilane (TEOS) were purchased from Sigma-Aldrich. All chemicals were purchased from Sigma-Aldrich unless otherwise stated. Spherical bioactive glass nanoparticles containing 85SiO_2_/15CaO (mol%) were manufactured through a base-catalyzed sol-gel approach according to a previously reported method [20]. Briefly, 1 g of calcium nitrate and 10 g of P123 were dissolved in 150 mL of solution consisting of 100 mL of distilled water (DW), 25 mL of ethanol, and 25 mL of ammonia solution (28%) at 50 °C under mechanical stirring at 500 rotations per minute (rpm) for 30 min. NH_4_OH was added to adjust the pH to 12.5, and then 1 g of Ca(NO_3_)_2_·4H_2_O was dissolved in the above solution. In a separate vessel, 5.35 mL of TEOS was added to 40 mL of absolute methanol, and this mixture was slowly added to the solution prepared above with the application of a sonoreactor (LH700S ultra-sonic power generators, Ulsso Hitech, Cheongwon-si, Korea) at 20 kHz and 700 W for 30 min. After maintaining mechanical stirring overnight, the white BGns were centrifuged at 5000 rpm and washed successively with DW, acetone, and 70% ethanol. The precipitate was dried at 70 °C overnight and finally heat-treated at 600 °C for 5 h with a constant heating rate (1°/min).

Scanning electron microscopy (SEM; Sigma300, ZEISS, Oberkochen, Germany) at 15 kV was used to examine the surface morphology of the BGns. The bioactive glass nanoparticles were analyzed by X-ray diffraction (XRD, Rigaku, Ultima IV, Tokyo, Japan) using CuKα radiation at 40 kV and 40 mA, and two theta (2θ) values were obtained from 5° to 80° with a step size of 0.01°. The elemental composition was determined by energy-dispersive spectroscopy (EDS; Quantax EDS, Bruker, Berlin, Germany).

### 2.2. Formulation and Characterization of Pre-Mixed-RCS@BGn

Three different groups of pre-mixed-RCS@BGn were formulated according to Figure 1C: BGn 0%, BGn 0.5% and BGn 1%. The commercial root canal sealer Bright Endo MTA Sealer (MTA10, 21K10-01; Genoss, Suwon, Gyeonggi-do, Korea) was used as the control and named BGn 0%. BGns were added to pre-mixed-RCS at 0.5 weight% for BGn 0.5% and at 1 wt% for BGn 1%. The setting time was too short for sealers with more than 1% BGn.

All BGn groups were molded into discs with a diameter of 15 mm and a height of 4 mm and placed at 37 °C under 5% CO_2_ with 95% humidity for 72 h. The sizes and shapes were observed using scanning electron microscopy (SEM) at 10 kV. X-ray diffraction (XRD) was used to study the structural properties, and the diffractometer was operated at 40 kV with a current of 40 mA and a scan rate of 2θ/min with a 2θ range of 5–80°. The elements were analyzed by energy dispersive X-ray spectroscopy (EDS) at 30 kV.

### 2.3. Physical Properties of Pre-Mixed-RCS@BGn

To measure the flowability of each group, 0.05 mL of sealer was placed evenly on a plate, creating a circle. Approximately 180 s after mixing began, a second plate was placed on top of the sealer. The sealer was left to flow for 10 min under a 120 g weight. The flow distance was measured using a Vernier caliper (series 530; Mitutoyo, Tokyo, Japan). The smallest and largest diameters of the sealer in the compressed disc were measured and averaged as the representative value with a sample size of 3 in each group.

Film thickness was measured according to the procedures in the ISO 6876 standard [21]. Two flat surfaced square glass plates with a thickness of 5 mm and an area of 200 mm^2^ were prepared. The film thicknesses were measured with a digital micrometer (Absolute Digimatic 500-197, Mitutoyo Corp, Kawasaki, Japan). After measuring two flat glasses together, 0.015 g of sealer was placed between the plates. The glass plates with sealer were vertically pressed with a load of 150 N for at least 10 min, and then the thicknesses of the glass plates with the spread sealer were measured. The film thickness of each sample was considered to be the difference between the thickness measurements before and after loading. Each specimen had a sample size of 3, and the mean film thickness values were calculated.

The setting time was measured in accordance with the standards for premixed root canal sealing materials (ISO 6876, *n* = 3). A stainless-steel ring mold (r = 10 mm, d = 2 mm) was filled with sealer and placed on a metal block conditioned at 37 ± 1 °C in a cabinet for at least 1 h. As the setting time stated by the manufacturer approached, a Gilmore-type indenter needle was applied vertically to the horizontal surface of the specimens. The time until indentations ceased to be visible was independently recorded as the setting time (*n* = 3).

To measure solubility, two specimens were placed in a shallow dish (Petri or other suitable glass or porcelain having a diameter of approximately 90 mm, with a minimum volume of 70 mL and of known mass to the nearest 0.001 g) so that they did not touch and remained undisturbed. Water (50 ± 1 mL) was added, and the covered dish was placed in a cabinet for 24 h. The specimens were washed with 2–3 mL of fresh water and then removed. After the water was evaporated from the dish without boiling, the dish was dried to constant mass at 110 ± 2 °C, cooling the dish in a desiccator to room temperature before each weighing (accurate to the nearest 0.001 g).

Molds with a diameter of 15 mm and a thickness of 4 mm were used to measure radiopacity. The sealer in the mold was pressed with the cover on the top and bottom to generate a 1 mm thick specimen. The specimen was positioned in the center of the X-ray film (Kodak Insight, Rochester, NY, USA), adjacent to the step wedge of 1–9 mm, which was scaled 1 mm apart. The specimen, step wedge, and film were irradiated with X-rays using a Kodak-2200 X-ray irradiator (Kodak Insight, Rochester, NY, USA), at 70 kV and 7 mA, at a target film distance of 300 mm and at an exposure time of 30 s. The digital X-ray images were collected and investigated with a gray level analysis program (ImageJ version 1.53a, National Institutes of Health, Bethesda, MD, USA). The density of the specimen’s image and the aluminum step wedge were compared using the optical density instrument. The radiopacity equivalent of the specimens was expressed in mm of aluminum. Each sample was tested 3 times, and the calculation from previous studies [22] was used to convert the optical density of the sample into the corresponding aluminum thickness. If the numerical value of the optical density of the image of the specimen was less than the density of the 3 mm aluminum step, the sealer was determined to have a radiopacity greater than 3 mm of aluminum.

### 2.4. Chemical Properties of Pre-Mixed-RCS@BGn

The surface contact angle of each sample was measured to determine hydrophilicity. A Phoenix contact angle measurement system (PHX300, SEO, Suwon, Korea) was used with the sessile drop method. Each sealer specimen was prepared with a disc 15 mm in diameter and 4 mm in height, and approximately 10 μL of distilled water was dropped on top of each specimen disc. Each droplet on the sealer sample disc was measured 10 s after dropping and recorded automatically with a video recording device. According to the video data, the surface contact angles were analyzed with a Phoenix 300 touch automatic contact angle analyzer (Surface Electro Optics Co., Suwon, Korea) (*n* = 3).

One gram of each mixed sealer sample was molded into a disc with a diameter of 15 mm and a height of 4 mm and set at 37 °C under 100% humidity for one hour. The set materials were then immersed in 5 mL of distilled water for 4 h and 1, 3, and 4 days, and the solutions were used for identifying ion release (*n* = 3). The released liquid was filtered through a 0.2 μm filter, and 1 mL of the filtered solution was transferred to a test tube that contained 10 μL of concentrated nitric acid (HNO_3_; Wako Pure Chemical Industries, Osaka, Japan). The pH of the extract using culture media was measured using a digital pH meter (Orion 4 Star, Thermo Scientific Pierce, Rockford, IL, USA). Three measurements were performed on each sample, and all analyses were independently performed in triplicate to confirm reproducibility. The concentrations of calcium and silicon ions were then analyzed by inductively coupled plasma atomic emission spectrometry (ICP–AES; OPTIMA 8300, PerkinElmer, Waltham, MA, USA).

### 2.5. Antibacterial Ability

*Enterococcus faecalis* (ATCC 19433) was purchased from the American Type Culture Collection (ATCC; Manassas, VA, USA). All bacterial extracts were kept in glycerol stock solution in a −80 °C deep freezer. The extracts were streaked on plates with tryptic soy agar (TSA; Difco Laboratories, Becton Dickinson, Sparks, MD, USA) and incubated at 35 ± 2 °C for 18 h. After incubation, a single colony was transferred to tryptic soy broth (TSB; Difco Laboratories, Sparks, MD, USA) and incubated with shaking. All extracts were routinely grown in their respective culture medium for 1 h under aerobic conditions at 35 ± 2 °C for 18 h to produce 2 × 10^6^ CFUs (colony forming units)/mL. Then, 100% concentrated and 50% diluted extracts were prepared (250 μL) and mixed with 250 μL of *E. faecalis* (2 × 10^6^ CFUs/mL). Next, the bacterial extracts were treated in a shaking incubator for 1 h. Finally, 50 μL of PrestoBlue was added. The extracts were observed under a light microscope, and the optical density (OD) was measured at 570~600 nm at 30 min intervals using a microplate reader (BioTek, Winooski, VT, USA) (*n* = 5). The plate counting method was applied in agar medium to observe live bacteria more accurately.

### 2.6. Cell Viability

Cell viability was assessed according to ISO 10093-5:2009 [23]. Human bone marrow-derived mesenchymal stem cells were placed in a 96-well plate with 100 μL of medium, incubated for 24 h, and then washed with phosphate-buffered solution (PBS; Gibco, Grand Island, NY, USA). Diluted extracts (0, 6.25, 12.5, 25, 50, 100%) of the 3 test groups were added for 24 h of incubation based on the MTS assay (CellTiter 96 Aqueous One Solution Cell Proliferation Assay, Promega, Madison, WI, USA). Human bone marrow-derived mesenchymal stem cells cultured in growth medium were used as a negative control. The optical density (OD) was measured with a microplate reader (SpectraMax M2e, Molecular Devices, Sunnyvale, CA, USA)/iMark microplate reader (Bio-Rad, Hercules, CA, USA) at a wavelength of 490 nm (*n* = 5). A live/dead assay was conducted by staining samples with 0.15 mM calcein AM and 2 mM ethidium homodimer-1 for 45 min to evaluate cell viability. The fluorescently stained live (green) and dead (red) cells were visualized by confocal laser scanning microscopy (CLSM; LSM700, Carl Zeiss, Thornwood, NY, USA).

### 2.7. Osteogenic Differentiation Assay

Human bone marrow-derived mesenchymal stem cells (hMSCs) were seeded in 24-well plates at a density of 1 × 10^4^ cells/well, and after 24 h, the cells were added to pre-mixed RCS@BGn. The noncytotoxic 25%, 12.5% and 6.25% diluted extracts were added for culture, followed by an alkaline phosphatase (ALP) activity assay and alizarin red S (ARS) staining. Osteogenic differentiation medium was added for 7 and 14 days. ALP staining was conducted by using a staining kit (Sigma-Aldrich, St. Louis, MO, USA). The cells and 25%, 12.5%, and 6.25% diluted extracts were washed with phosphate-buffered saline (PBS; Tech and Innovation, Chuncheon, Korea) twice and fixed with 4% paraformaldehyde (PFA) at room temperature for 30 min. The fixed cells were washed with PBS three times and stained with ALP solution dissolved in distilled water for 1 h at 37 °C. The stained cells were rinsed with PBS three times and examined under an optical microscope. After staining, quantification was performed using ImageJ software (Bethesda, MD, USA). Alizarin red S (ARS, Sigma-Aldrich, USA) staining was also performed on hMSCs. hMSCs were seeded in 24-well plates at a density of 1 × 10^4^ cells/well in osteogenic differentiation medium for 14 days. The cells were washed twice with PBS and then fixed with 4% PFA for 30 min. The fixed cells were also washed 3 times with PBS and subsequently stained with 40 mM ARS solution for 10 min. After staining, the cells were washed five times with distilled water, and optical images were obtained using a microscope. The stained calcium deposits were dissolved in 10% (*w*/*v*) cetylpyridinium chloride (CPC; Sigma-Aldrich, MO, USA) solution on a rocking shaker. After 1 h, the released solutions were transferred to 96-well plates and the absorbance values were measured at 562 nm using a microplate reader.

### 2.8. Statistics

All data are represented as the mean ± SD of independent triplicate experiments. Images are representative of at least triplicate independent experimental sets. Statistical analysis was carried out using one-way analysis of variance (ANOVA) followed by Tukey’s post hoc test with a significance level of 0.05 after performing the Shapiro–Wilk test to confirm normality.

## 3. Results and Discussion

### 3.1. Formulation and Characterization of Pre-Mixed-RCS@BGn

BGns were synthesized by a sol-gel method to a diameter of 100 ± 20 nm and showed a spherical morphology in the SEM image (Figure 1A). In order to cross verify the SEM analysis, dynamic light scattering (DLS) method was used, and the diameter was 152 ± 25 nm. The observed morphology by SEM exhibited nanoparticles that were relatively homogeneous. The XRD pattern in Figure 1B of the as-prepared BGns showed a broad amorphous peak between 2θ = 15° and 35°, which is a typical characteristic of amorphous bioactive glasses [24]. The BGn powder had an elemental composition of Ca:Si = 15:85 in mol%, similar to previous studies [20,25,26]. These nanoparticles are expected to release therapeutic ions and promote the regeneration of periapical tissue [27].

Pre-mixed-RCS@BGn was fabricated by mixing pre-mixed-RCS and BGns (Figure 1C). BGn concentrations of 0.5% and 1% were chosen for sealer preparation (Figure 1D) after preliminary experiments using 1.5% and 2% BGns. The setting time was too short for sealers with more than 1% BGns, which can be explained by the greater water absorption during the setting reaction due to the presence of the nanoparticles [28].

Pre-mixed-RCS@BGn with 0%, 0.5%, and 1% BGns were prepared in solid disc form after setting, and their surface morphology and elemental proportions were analyzed by SEM, XRD and EDS. All sealers had rough surfaces with slightly different particle sizes in the SEM images (Figure 1E). The XRD results showed that the calcium silicate peaks were the same in the calcium silicate sealers with and without BGns (Figure 1F). EDS analysis of the original calcium silicate sealer confirmed the presence of calcium, oxygen, carbon, silicon, zirconium, and bismuth (Figure 1G). The original sealer had a composition of 38.62% calcium, 7.30% silicon, 7.48% carbon, and 37.31% oxygen after mass normalization. In the sealers incorporated with BGns, the mass percentage of calcium decreased while that of silicon increased as the concentration of the BGn additive increased (Figure 1H). The mass percentages of calcium were 38.30 ± 0.31%, 33.44 ± 0.69%, and 33.71 ± 0.72% in the 0%, 0.5%, and 1% groups, respectively, and the mass percentages of silicon were 7.25 ± 0.06%, 8.07 ± 0.09%, and 8.82 ± 0.67% in the 0%, 0.5%, and 1% groups, respectively.

The SEM surface morphologies of pre-mixed-RCS@BGn exhibited inhomogeneous microstructured surfaces with nanoparticles of different sizes (100 ± 20 nm), which can be explained by the XRD patterns. However, pre-mixed-RCS showed fewer particles, and their sizes were much larger (500 ± 100 nm). The XRD spectra exhibited peaks for calcium silicates, calcium hydroxide, and calcium silicate hydrate [19]. When calcium silicate sealers react with water after setting, crystalline calcium hydroxide and calcium silicate hydrate are formed, followed by hydroxyapatite formation [29]. Calcium hydroxide plays a role in maintaining alkaline conditions and has the ability to both eradicate microorganisms and promote the differentiation of undifferentiated stem cells in the pulp [30]. Calcium silicate hydrate fills the gap between dentin and the sealer and grows inside the dentinal tubules, obstructing the invasion of microbes [31]. The XRD peaks were similar in all groups, indicating that BGns did not affect the typical crystallinity of pre-mixed-RCS [32]. In addition, EDS analysis confirmed that more silicon ions were included in the BGns than calcium ions.

### 3.2. Physical Properties of Pre-mixed-RCS@BGn

The flowability, film thickness, setting time, solubility, and radiopacity were measured according to ISO standards (Figure 2A–F). According to ISO 6876, the flowability should be more than 17 mm, and the film thickness should be less than 50 μm; both values were satisfied by all groups of sealers with and without BGns. As more BGns were added, the flowability decreased, which refers to the increase in density and viscosity. The setting time of all the materials was within the range stated by the manufacturer (60 min), and it seemed to decrease slightly as more BGns were added. The solubility was less than 3% by mass, and the aluminum thickness was more than 3 mm in the sealers with and without BGns, which conformed to the ISO standards. Chemical composition and the physical properties of pre-mixed-RCS@BGn were summarized in Table 1.

### 3.3. Chemical Properties of Pre-Mixed-RCS@BGn

All the materials had water contact angles from 10 to 35 degrees, and the angle decreased slightly as more BGns were added (Figure 3A). The water contact angle data showed that all sealers exhibited similar hydrophilicity, which is related to sealing ability and antimicrobial activity [33]. Good wettability enables the penetration of sealing materials into the microirregularities by enhancing the adhesion between a sealer and the root canal wall [34].

The pH values of the extracts from the 0%, 0.5%, and 1% BGn incorporated sealers were measured and found to be between 9 and 13 (Figure 3B). For all of the sealers, the pH values increased until 24 h and then decreased after 24 h. From the 24-h extract, the pH value of the BGn 1% group was slightly lower than those of the control and BGn 0.5% groups. A much lower pH value in the BGn 1% group was observed in the 168-h extract. Moreover, these results confirm that calcium silicate sealers produce hydrated calcium silicate hydrates and calcium hydroxide, which release hydroxyl ions and provide an alkaline environment [35]. All groups maintained high alkalinity to produce a hostile environment for the survival and proliferation of bacteria [36].

Ca and Si ions are the core components of calcium silicate sealers and bioactive glass nanoparticles and are released in all extracts (Figure 3C). The number of Ca ions released increased from 4 to 168 h in the BGn 0% group and the BGn 0.5% group, while there was a slight decrease from 24 to 72 h but an overall increase in the BGn 1% group. The reason why there is discrepancy of Ca and Si ion tendencies might be the deposition of each ion into sealer, which is helpful for enhancing acellular biomineralization on sealers. BGns are known to undergo ionic dissolution when immersed in SBF, including calcium ion. Then, silanols and hydrated silica gels are formed, which can explain for the decrease of Si ions.

From the 24-h extract, the concentration of Ca ions decreased as more BGns were added. Furthermore, the number of Si ions released decreased from 4 to 168 h in all sealers. BGns are known to undergo ionic dissolution when immersed in SBF, including calcium ions [37,38]. Silanols and then hydrated silica gels are formed, which can explain the decrease of Si ions [39].

In the 24-h extract, there was a significant increase in Si ions in pre-mixed-RCS@BGn compared to the control. The tendency toward ion release was closely related to the composition of the BGns, as supported by previous studies [40]. Many studies have explored the effects of nanoparticles incorporating calcium ions for their osteoconductivity and mineralization properties [41]. However, our study makes a significant contribution by explaining the effect of BGns with more silicon ions. Nanoparticles with silicon or silica ions have also been reported to have a crucial role in bone regeneration, which supports our BGn content [42].

### 3.4. Antibacterial Activity of Pre-Mixed-RCS@BGn

The effects of sealers with BGns on the viability of *Enterococcus faecalis* were tested in vitro. The antibacterial activity was tested using leaching of sealers with and without BGns (Figure 4A). The optical density of the microbe was lower in the 100% extracts than in the 50% extracts. Furthermore, the density of the bacteria decreased slightly as more BGns were added, but in a negligible range. The CFUs of *Enterococcus faecalis* in the 100% extract of each sealer were also evaluated using a microplate reader (Figure 4B). Antibacterial activity was found in all groups of sealers, and the bactericidal effect increased trivially as more BGns were incorporated. The transmission electron microscopy images (Figure 4B) corresponded to these results.

*Enterococcus faecalis* has been implicated as a persistent species found in root canal infections that is commonly recovered from root canals with posttreatment disease [43,44]. *E. faecalis* adheres to the root canal and invades the dentinal tubules well due to the presence of a serine protease and collagen-binding protein [45]. Therefore, the germ-destroying effects of calcium silicate-based MTA sealers have been investigated in multiple studies [7,46,47]. In the current study, sealers with BGns maintained their antibacterial properties. The 24-h extract of all sealers had a high pH between 11 and 12.5, which was consistent with a previous study stating that the growth of *Enterococcus faecalis* was hampered at pH values between 10.5 and 11.0, and the bacteria were unable to withstand at pH values higher than 11.5 [48]. The influence of pH on antimicrobial activity can be explained by Ca(OH)_2_, which releases hydroxyl ions in an aqueous environment. Hydroxyl ions act as free radicals that have the utmost reactivity with other biomolecules, affecting the cellular components of bacteria [49].

### 3.5. Cytocompatibility of hMSCs with Pre-Mixed-RCS@BGn

A cell viability test was conducted using the MTS assay to determine cytotoxicity to human mesenchymal cells (hMSCs) in Figure 5A. The cell viability results of the 50% sealer extracts were reevaluated by acquiring fluorescence microscopy images (Figure 5B). There was severe cytotoxicity for the 100% extract with less than 50% cell viability, and cell viability increased as the extract concentration decreased. Moreover, for all the extracts except for the 12.5% extract, cell viability increased as more BGns were incorporated. Fluorescence microscopy was used to observe the cytocompatibility. The BGn 1% group showed the highest cell viability, followed by the BGn 0.5% group and the BGn 0% group. Fewer live cells were detected in the 50% extract of the BGn 0% group than in the groups reinforced with BGns, confirming the results of the cell viability assay.

Human mesenchymal stem cells are known to be vulnerable to changes in pH and survive at pH levels between 6.6 and 7.8 [50]. A moderate intracellular pH serves as a permissive or obligatory signal for cell proliferation, which corresponds with the above results [51]. As more BGns were added, the pH levels decreased to become suitable for dental pulp stem cells while maintaining an alkaline environment. It is also assumed that a decrease in calcium ions may influence cell viability. Other studies have shown that the elevated concentrations of calcium ions increased the cellular apoptotic rate and resulted in more necrotic human dental pulp cells [52]. BGns with an appropriate amount of calcium could allow cell viability and show supreme mineralization ability.

### 3.6. Osteogenic Differentiation and Biomineralization of Pre-Mixed-RCS@BGn

The osteogenic potential of each root sealer was analyzed in vitro (Figure 6). The 6.25%, 12.5%, and 25% extracts were subjected to both alkaline phosphate (ALP) activity and alizarin red S (ARS) staining examinations because there was severe cytotoxicity in the abovementioned 50% and 100% extracts. ALP is an early marker of osteogenic differentiation [53], and the optical images showed a very dark violet color in the BGn 1% group treated with the 25% extract for 7 and 14 days (Figure 6A). In addition, calcium deposition in human mesenchymal stem cells cultured for 14 days was visualized by ARS staining to qualitatively compare the mineralization in each group (Figure 6B). The red color indicates that the calcium content was significantly denser in the BGn 1% group, followed by the BGn 0.5% and BGn 0% groups.

Therapeutic ions, such as calcium and silicon, are known to stimulate osteogenic differentiation [54]. The results here indicate that BGn incorporation influenced osteogenic differentiation by releasing more silicon or silica ions. Silicon is considered an essential element for metabolic processes and is involved in the formation and calcification of bone tissue [55]. Aqueous silicon ions are able to induce the precipitation of hydroxyapatite, the inorganic phase of human bone [56]. Other studies support that silicon ions prompt young bone formation by stimulating osteoblasts and promoting mineralization, even playing a role in dentin formation [57]. Furthermore, the underlying mechanism for osteogenic differentiation is the activation of gap junction communication and induction of gene expression patterns related to osteogenic differentiation, such as BMP-2 and BSP [58]. The role of silicon in vivo supports the overall enhanced osteogenic differentiation and biomineralization capacity of pre-mixed-RCS@BGn.

## 4. Conclusions

In conclusion, pre-mixed-RCS@BGn showed less cytotoxicity and induced more osteogenic differentiation at nontoxic concentrations than pre-mixed-RCS without BGns. Chemical property analysis showed differences in the pH changes and released Ca ions and Si ions, supporting the mentioned in vitro functionalities while maintaining appropriate properties as root canal sealers. Therefore, under the influence of the extracts at nontoxic concentrations, premixed MTA sealer seems to be a promising biomaterial that is helpful for cell viability near the apical foramen and for dental tissue regeneration. Understanding the mechanism of tissue regeneration will expand the experiments to genetic or protein level. Further in vivo studies will be necessary to evaluate biocompatibility within the general system, including in pulp tissue.

## Figures and Tables

**Figure 1 pharmaceutics-14-01903-f001:**
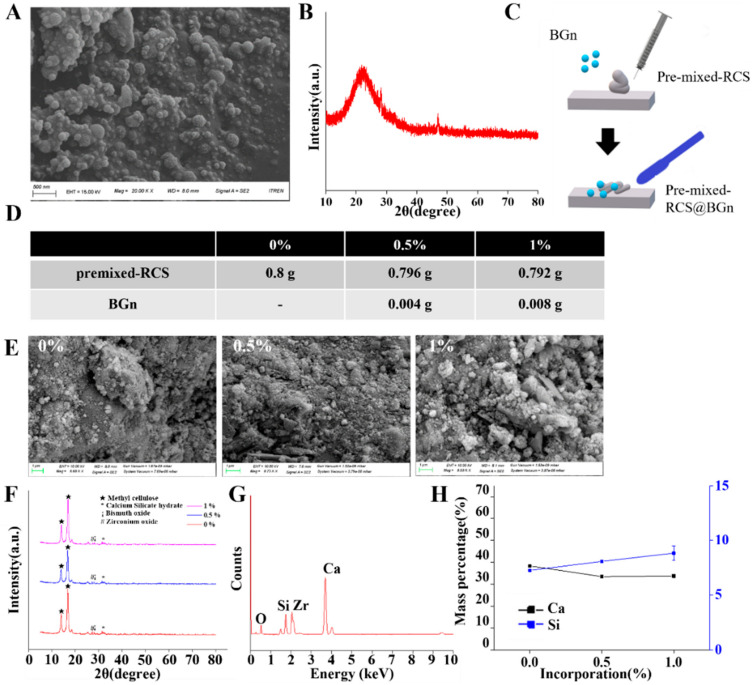
Characterization of BGns and pre-mixed-RCS@BGn. (**A**) SEM image of BGn powder confirmed that the powder was composed of nanoparticles. (**B**) The XRD pattern exhibited a broad amorphous peak, which is a typical characteristic of bioactive glasses. (**C**) Pre-mixed-RCS@BGn was formulated by mixing pre-mixed-RCS and BGns. (**D**) BGns (0%, 0.5%, 1% mass percentage) were incorporated into pre-mixed-RCS. (**E**) SEM image and XRD analysis of pre-mixed-RCS@BGn confirmed that pre-mixed-RCS and BGns were well agglomerated. (**F**) The XRD results showed that pre-mixed-RCS@BGn had 2 peaks between 10° and 20°, which revealed the bioactivity of the BGns. (**G**) The EDS analysis evaluated the composition of pre-mixed-RCS, which consists of calcium, oxygen, carbon, silicon, zirconium, and bismuth. (**H**) The amount of silicon increased as the concentration of the BGn additive increased, which was clarified by the EDS results.

**Figure 2 pharmaceutics-14-01903-f002:**
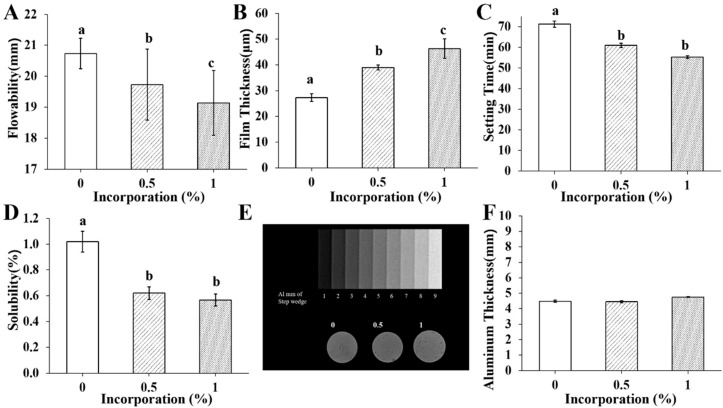
Physical properties of pre-mixed-RCS@BGn. (**A**) Although the flowability decreased as the concentration of BGns increased, all materials met the ISO standard (17 mm, *n* = 3). (**B**) Film thickness increased as the concentration of BGns increased, while all materials presented film thicknesses less than 50 µm, which met the ISO flowability standard (*n* = 3). (**C**) The setting time was within the range stated by the manufacturer (60 min). (**D**) The solubility of the set sealers did not exceed 3% by mass. (**E**) Radiographic image of the radiopacities of the BGN 0%, 0.5%, and 1% groups and their equivalence to those of the aluminum step wedge. (**F**) The set sealers have a radiopacity equivalent to that not less than 3 mm of aluminum, which meets the ISO radiopacity standard (*n* = 3). Different letters for each group indicate significant differences between them (*p* < 0.05).

**Figure 3 pharmaceutics-14-01903-f003:**
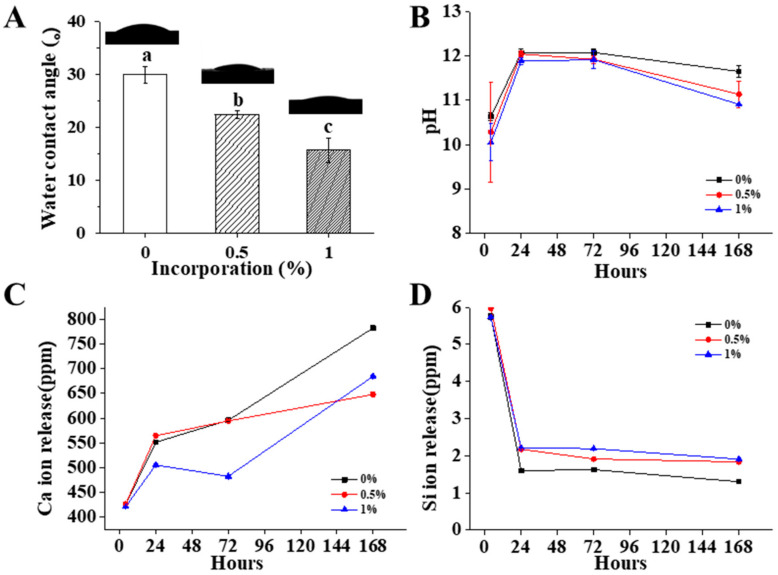
Chemical properties of pre-mixed-RCS@BGn. (**A**) The water contact angle of each group showed hydrophilicity (15–30 degrees) (*n* = 3). (**B**) pH changes over 7 days (*n* = 3). There was no significant difference in pH between the three groups, and all materials were in an alkaline environment. (**C**,**D**) Ion release over 7 days (*n* = 3). Calcium ion release did not show a significant difference, while silicate ion release increased as the BGn concentration increased. Different letters for each group indicate significant differences between them (*p* < 0.05).

**Figure 4 pharmaceutics-14-01903-f004:**
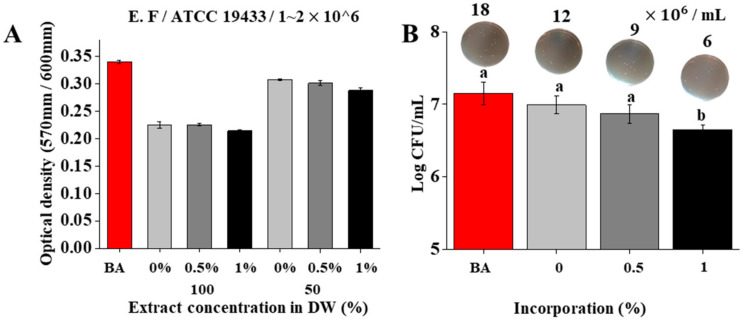
Antibacterial effects of the bioactive set sealers analyzed with *Enterococcus faecalis*, 2 × 10^6^ CFUs (colony forming units)/mL. (**A**) Optical density of *Enterococcus faecalis* cultured media. *Enterococcus faecalis* cultured with sealer extract media (100%) exhibited higher antibacterial activity compared to sealer extract media (50%). (**B**) Log CFUs (colony forming units)/mL of *Enterococcus faecalis* cultured with sealer extract media (100%) combined with 0, 0.5, and 1% BGns. (18, 12, 9, 6) × 10^6^ are the CFUs (colony forming units)/mL value of *Enterococcus faecalis*. As the concentration of BGn increased, the number of *Enterococcus faecalis* decreased. Different letters for each extract indicate significant differences between them (*p* < 0.05).

**Figure 5 pharmaceutics-14-01903-f005:**
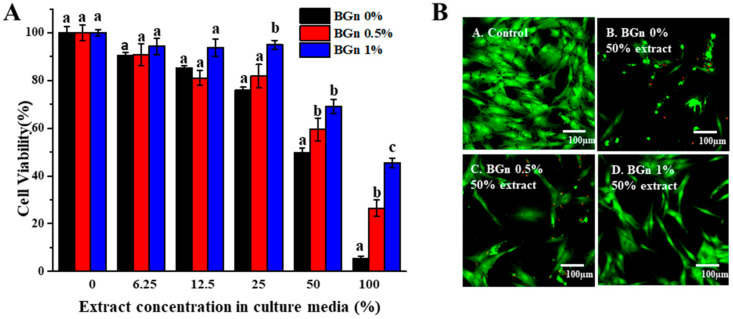
Cytocompatibility of hMSCs with premixed-RCS@BGn. (**A**) Cell viability results at various concentrations of sealer extracts with 0.5 and 1% BGn. Cell viability increased as the concentration of BGns increased. Different letters for each extract indicate significant differences between them (*p* < 0.05). (**B**) Live and dead cells exposed to 50% sealer extracts with 0.5 and 1% BGns. Live (green) and dead (red) cells were observed by fluorescence microscopy. Cell viability decreased when the extract concentration increased.

**Figure 6 pharmaceutics-14-01903-f006:**
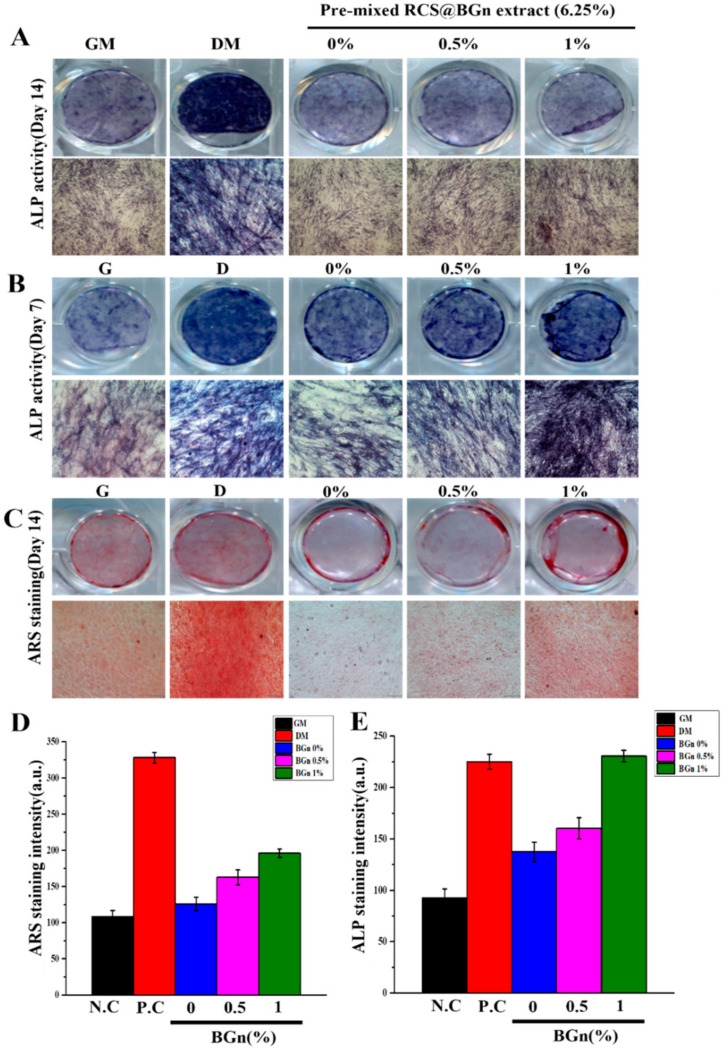
Osteogenic differentiation and biomineralization. Optical images of (**A**) alkaline phosphatase (ALP) activity on Day 7, (**B**) alkaline phosphatase (ALP) activity on Day 14, and (**C**) alizarin red S (ARS) staining on Day 14 for the 6.25% extracts 6.25% BGn 0%, BGn 0.5%, and BGn 1%. Negative (hMSCs cultured in growth medium) and positive (hMSCs cultured in differentiation medium) controls were used. Active osteogenic differentiation occurred in BGn 1%, displaying a significant dark blue color for ALP and redder ARS staining using hMSCs compared to those of the other groups (*n* = 3). Quantitative analysis of (**D**) alkaline phosphatase (ALP) activity on Day 14 and (**E**) alizarine red S (ARS) staining on Day 14.

**Table 1 pharmaceutics-14-01903-t001:** Chemical composition and physical properties of pre-mixed-RCS@BGn. Different letters for each group indicate significant differences between them (*p* < 0.05).

	Pre-Mixed-RCS + 0% BGn	Pre-Mixed-RCS + 0.5% BGn	Pre-Mixed-RCS + 1% BGn
**Chemical** **composition (%)**	Ca: 38.30 (0.31) Si: 7.25 (0.06)	Ca: 33.44 (0.69) Si: 8.07 (0.09)	Ca: 33.71 (0.72) Si: 8.82 (0.67)
**Flowability (mm)**	20.74 (0.49) ^a^	19.73 (1.15) ^b^	19.14 (1.05) ^c^
**Film thickness (µm)**	27.33 (1.53) ^a^	39.00 (1.00) ^b^	46.33 (3.79) ^c^
**Setting time (min)**	71.33 (1.53) ^a^	61.00 (1.00) ^b^	55.33 (0.58) ^b^
**Solubility (%)**	1.02 (0.08) ^a^	0.62 (0.05) ^b^	0.57 (0.05) ^b^
**Aluminum** **Thickness (mm)**	4.49 (0.06)	4.45 (0.07)	4.75 (0.03)
**Water contact angle (°)**	29.94 (1.53) ^a^	22.41 (0.76) ^b^	15.68 (2.33) ^c^

## Data Availability

Not applicable.

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
