# Peer review of "Premixed Calcium Silicate-Based Root Canal Sealer Reinforced with Bioactive Glass Nanoparticles to Improve Biological Properties"

_pharmaceutics, 2022, doi:10.3390/pharmaceutics14091903_

Round 1
Reviewer 1 Report
In the present study, investigators tried to reinforce a premixed calcium silicate root canal sealer with BGns to improve biological properties. After reviewing the manuscript, I have the following questions:
1. The bioactivity and osteogenic properties of BG are closely related to its chemical composition and morphology. Has the BG chosen in this study been studied? Please add relevant information.
2. Authors concluded that pre-mixed-RCS@BGn showed less cytotoxicity and induced more osteogenic differentiation at nontoxic concentrations than pre-mixed-RCS without BGns. But do not think present results are strong enough to support this conclusion. Some suggestions as following: Cell viability results need statistical analysis. In addition to the qualitative analysis of ALP and ARS, the osteogenic potential of each root sealer needs to be proved by quantitative experiments, such as osteogenic related genes, proteins and mineralized.
3. Physical and chemical properties of pre-mixed-RCS@BGn,including flowability, film thickness, setting time, solubility and hydrophilicity have been shown a decrease.
4. The release of ions of Si and Ca in tissue fluid and formation of HA are important reasons for the bioactivity and osteogenesis of BG. The results of Fig3C and D were confusing. It is recommended to repeat the experiment. Usually, the ion release experiment of BG is to soak the material in SBF to simulate the environment in vivo.
5. Fig 1A,E, F are unclear and cannot be identified.
6. Fig1H, Line charts are not properly for the data.
Author Response
Thank you for your comment. Please see the attachment.

Reviewer 2 Report
Dear Authors,
I read your manuscript, that aims to evaluate the physical, chemical, and biological properties of premixed calcium silicate root canal sealer reinforced with bioactive glass. The topic is interesting, but some issues need clarification.
Best regards,

Author Response

(The authors gave the same response as above.)

Reviewer 3 Report
In this study, authors evaluate the physical, chemical, and biological properties of premixed calcium silicate root canal sealer reinforced with BGns (pre-mixed-RCS@BGn). The physical properties such as flowability, film thickness, setting time, solubility and radiopacity were measured, and chemical properties such as pH and ion release were examined. Then, cellular bioactivity in terms of cytocompatibility and osteogenic differentiation were tested. The results are interesting, but there is suggestion to improve the paper. My comments are appended below
Can authors provide the chemical composition and Physical properties (e.g. Flow, film thickness, setting time, %Solubility, Radiopacity) of bioactive glass-based sealers in a tabular form to make it easy to understand?
Check abbreviation ´bioactive glass nanoparticle (BGn)´ as they are mentioned twice as BGns and BGn, follow a single consistent acronym in order to not confuse the readers.
Cite a latest report DOI: 10.2174/1381612822666151210124001 with line 78-79 sentence ending with ´could enhance the physical, 78 chemical, and biological properties of root canal sealers´ to make reference list up to date.
Introduction line 81, change ´cytologic´ to ´biocompatibility´.
Martials and methods
Provide catalogue number, suppler, country/region for each chemical us din the study.
The figures as figure 1.C description comes before figure 1.A-B. follow same to check other figure cited in the main text.
Line 135, provide a suitable reference for the sentence ´Film thickness was measured according to the procedures in the ISO 6876 standard´. Also, in line 214.
Figure 1.G-H labels are blurry and hard to read, please improve. In fig. 1. (F) The XRD results, label peak position and fig. 1. (G) The EDS analysis, label signature elements according to peaks shown.
Figure 2, what are a,b,c in small latter over each data points, either label in figure or add in figure captions.
Did author recorded size distribution of nanoparticle using standard optical methods such as dynamic light scattering (DLS) to cross verify the SEM based nanoparticulate analysis?
Author Response

(The authors gave the same response as above.)

Round 2
Reviewer 1 Report
In response to previous review comments, the authors have made appropriate changes or discussions in this manuscript. No further comments.
Reviewer 3 Report
accept